# Analysis and Performance Evaluation of BDS-3 Code Ranging Accuracy Based on Raw IF Data from a Zero-Baseline Experiment

**Yu Liu [1], Fan Gao [1,2,\*], Junxiang Li [1], Yunqiao He [1], Baojiao Ning [1], Yang Liu [1], Sijia Chen [1] and Yanqing Qiu [1]**

[1] School of Space Science and Physics, Shandong University, Weihai 264209, China; 201900810123@mail.sdu.edu.cn (Y.L.); 201900810112@mail.sdu.edu.cn (J.L.); heyunqiao@mail.sdu.edu.cn (Y.H.); 202017724@mail.sdu.edu.cn (B.N.); 202000810242@mail.sdu.edu.cn (Y.L.); 201900800335@mail.sdu.edu.cn (S.C.); 201900830050@mail.sdu.edu.cn (Y.Q.)
[2] Institute of Space Science, Shandong University, Weihai 264209, China
\* Correspondence: gaofan@sdu.edu.cn

**Abstract:** China's BDS-3 global navigation satellite system has been built and is providing official open Positioning, Navigation, and Timing (PNT) service with full operational capability (FOC) since July 2020. The main new civil B1C and B2a ranging code signals are broadcasted on the two carriers with central frequencies of 1575.42 MHz and 1176.45 MHz, which were shared by other GNSSs. Compared with traditional signals, such as GPS L1 C/A and BDS B1I, the new civil signals have better modulation and wider bandwidth to be expected to achieve a better range performance. In order to evaluate code ranging accuracies directly, a zero-baseline experiment using a geodetic GNSS antenna and a four-channel intermediate frequency (IF) signal recorder was conducted. Two channels were used to receive the signals with a central frequency of 1575.42 MHz at a 62 MHz sampling rate, and the other two channels are for 1176.45 MHz. The raw IF data were post-processed using a software-defined receiver (SDR) to compute the code signal path differences between two channels with the same frequencies. Compared with the traditional hardware receiver, SDR has the characteristics of flexible use and good operability, but its running speed is slow. The root-mean-square (RMS) and bias values of the path differences from BDS B1C, BDS B2a, and GPS L5C were used to evaluate their accuracies. The results show that there is a weak negative correlation between the satellite elevation and the ranging accuracy when the satellite elevation ranges from 30° to 90°. The ranging accuracy of the B1C signal is lower than that of B2a, which may be caused by different code rates, bandwidth, and signal structure. The GPS L5C is used for precision analysis as a comparison. It shows that the code signal path differences accuracy of L5C is close to the B2a.

**Keywords:** BDS-3; ranging code; zero-baseline; raw IF data

## 1. Introduction

Since the 1980s, China has been committed to the development of its own global satellite navigation system with independent intellectual property rights, which was named the BeiDou Navigation Satellite System (BDS). The construction of the BDS is divided into three stages: the BeiDou double satellites positioning system (BDS-1), the BeiDou regional navigation system (BDS-2), and the global system (BDS-3) [1]. BDS-1 was announced in mid-2003 to operate two geostationary satellites and a backup satellite [2], which has now been deactivated. BDS-2 has officially started to provide Positioning, Navigation, and Timing (PNT) services to the Asia Pacific region on 27 December 2012 [3]. The BDS-3 was put into use after the successful launch of the last networking satellite in June 2020. After that, the number of BDS satellites that can provide navigation signals in space reached 55, including 30 networking satellites of BDS-3 [4]. Among the 30 networking satellites of BDS-3, there are 24 medium earth orbit satellites (MEO), three inclined geosynchronous

orbit satellites (IGSO), and three geostationary orbit satellites (GEO) [1–3]. It can provide global users with PNT services, short message communication, and other services, transmitting five public service signals centered at B1I (1561.098 MHz), B3I (1268.52 MHz), B1C (1575.42 MHz), B2a (1176.45 MHz), and B2b (1207.14 MHz) [5–9]. Compared with BDS-2, BDS-3 has significantly improved system coverage, spatial signal accuracy, spatial signal continuity and availability [10–12]. The continuity and availability of BDS-3 space signal are about 99.99% and 99.78%, respectively [13]. The signal transmitted at each frequency point is composed of ranging code and navigation data modulated on the carrier signal. Users can compute the propagation time of the signal at each time to calculate their position. According to the public system documents, the positioning accuracy of the Beidou satellite navigation system in the world will reach 10 m horizontally and 10 m vertically (95%). In the Asia-Pacific region, the positioning accuracies are 5 m horizontally and 5 m vertically (95%) [3].

In addition to the BeiDou navigation system, the GNSS has another three major global navigation systems. The Global Positioning System (GPS) was launched by the US Department of Defense in 1973. Its space segment consists of 24 satellites and 3 standby satellites. [14]. GPS III, a new generation of modern GPS satellites, has higher navigation accuracy, transmission power, and signal integrity with the new L1C civil signal [15]. The Galileo system consists of 24 operational satellites and 6 standby satellites [14]. The Galileo system plans to carry out extensive infrastructure development and deployment activities to achieve full operational capability (FOC) [16]. The GLONASS system launched its first networking satellite in 1982. Its fully operational constellation consists of 24 satellites distributed in three orbital planes. It will deploy GLONASS-K2 satellite and improve the tracking network to meet the needs of modernization [17].

With the advent and wide application of BDS-3, people have carried out all-round research on BDS-3, including signal characteristics, positioning accuracy, precision point positioning (PPP), etc. The comparison of BDS-3 features with other satellite navigation systems has also been widely promoted. Xie et al. [18] points out that the signal strength of the same type of satellite BDS-3 is greater than that of BDS-2. The MP combination is used to study the deviation between the pseudo-range and carrier phase measurement of BDS-3 and BDS-2. The authors point out that the BDS-2 coding is affected by the satellite-induced deviation of 1 m from the horizon to the zenith. Liu et al. [19] calculated the signal-in-space range error (SISRE) of different types of navigation satellites. The average SISREs of BDS-3 MEO, IGSO, and GEO satellites are 0.52 m, 0.90 m, and 1.15 m. The SISRE of BDS-3 MEO satellite is slightly lower than that of Galileo by 0.4 m, slightly higher than that of GPS by 0.59 m, and significantly better than that of GLONASS by 2.33 m. BDS-3 can achieve significant positioning accuracy. Fan [20] uses the user equivalent range error (UERE) to describe the signal quality of BDS-3 satellite in polar and global conditions. The UERE of BDS-3 is in the range of 0.5–1.0 m. To eliminate the ionosphere delay, orbit error, multipath error, troposphere delay, and satellite clock error, the zero-baseline experiment can be used for analysis. It can also evaluate the ranging accuracy of the signal. Yang et al. [21], using zero-baseline inter-station single-difference (SD) and ultra-short-baseline SD methods determines that the measurement accuracy of the BDS-3 system code and carrier phase are 33 cm and 2 mm, respectively. Deng et al. [22] used three zero-baseline experiments to find that the noise of code phase measurement and carrier measurement of BDS-3 are improved compared with BDS-2, and millimeter-level noise is measured on the B3I signal. Roberts et al. [23] compared and fused the observation accuracy of BeiDou and GPS by using the zero-baseline experimental data. The relative positioning accuracy of BeiDou can reach millimeter level, but it is lower than that of GPS. The integrated use of BeiDou and GPS can improve positioning accuracy. In addition, the initial positioning performance of the BDS-3 under different modes such as single-point positioning and RTK is also studied [24–26]. The above research on ranging performance and accuracy often requires a long time of complex observation and tedious post-processing of data operation, so it is impossible to directly obtain an index to quantitatively measure the signal ranging performance.

In this contribution, the ranging code accuracy and performance of two new BDS-3 civil signals B2a and B1C are preliminarily analyzed and compared with GPS L5C. The four-channel IF signal recorder is used by us to collect the raw IF signals of two BDS frequency points and conduct the zero-baseline experiment. The code ranging accuracy is analyzed by calculating the code signal path difference based on the waveform's extreme value difference, which can directly reflect the ranging performance, avoiding the complicated observation and data processing. This ranging accuracy evaluation method is first applied to the ranging accuracy analysis of BDS-3 B1C and B2a.

A software-defined receiver (SDR) was used to process the raw intermediate frequency (IF) signal to calculate the code-level path differences. SDR, which is a general framework for highly flexible multiple system solutions, is also applicable to general GNSS equipment [27]. It realizes all signal processing processes through programmable software or microprocessors Its advantage is particularly important in the case of GNSS because of the series of signals used in PNT services. If traditional hardware receivers are used, new hardware components must be developed or purchased, while SDR can include the processing function of new signals only by updating the software [28]. Our research covers multiple signal frequency bands, so SDR is selected for signal processing. However, although SDR has greater flexibility, it has an obvious disadvantage that the signal processing speed is extremely slow. The amount of raw IF signal data used in this study is large.

The remaining chapters of this paper are arranged as follows. In Section 2, the two civil signals of BDS-3 and other relevant details with the rationale of the zero-baseline experiment are described. In Section 3, the specific steps of our zero-baseline experiment will be described. In Section 4, we discuss the sampling results and the accuracy comparison among the three signal frequency points in detail. Finally, the conclusions and summary of the study, as well as the unsolved problems of this research, are discussed in Section 5.

## 2. Materials and Methods

### 2.1. Signals

BDS-3 has been provided official open PNT service since 31 July 2020. The BDS-2 regional satellite navigation system was built earlier than BDS-3, which has opened three civil signals for B1I (1561.098 MHz), B2I (1207.14 MHz), and B3I (1268.52 MHz). The open civil signals of BDS-2 are mainly modulated by Binary Phase Shift Keying (BPSK). After the completion of BDS-3, the BDS-3 inherits the B1I and B3I signals of BDS-2, and the signal modulation methods of B1I and B3I are strictly consistent with BDS-2 to ensure the interoperability of system signals. BDS-3 began to adopt new civil signal frequency points such as B1C (1575.42 MHz), B2a (1176.45 MHz), and B2b (1207.14 MHz) [5–9], and the observational accuracy of the BDS-3 signals is comparable to GPS L1/L2/L5 and Galileo E1/E5a/E5b, according to the report [29]. Frequency multiplexing exists among GNSSs. Figure 1 shows the frequency distribution of the current four global satellite navigation systems. If the frequency band numbers in Figure 1 have the same position on the frequency coordinate axis, they have the same frequency center, and there is a potential possibility of interoperability between systems. Frequency overlap puts forward higher requirements for compatibility and interoperability among systems.

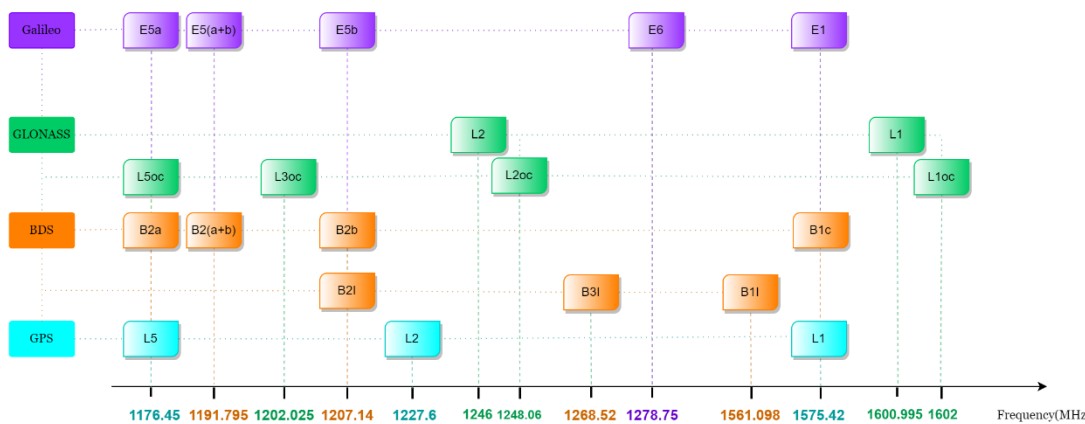

**Figure 1.** Frequency distribution of GNSSs.

B1C and B2a are the service signals that BDS-3 mainly opens to users. B1C and B2a can broadcast signals through 24 medium earth orbit satellites (MEO) and 3 inclined geosynchronous orbit satellites (IGSO). The signals broadcasted by the satellite are distinguished by capturing different pseudo-random noise codes (PRN). PRN numbers ranging from 19 to 46 belong to BDS-3 satellites, of which 38, 39, and 40 are IGSO satellites. The code length of the two signals is 10230, uniformly. However, the difference is that the coding rate of B2a is 10.23 Mcps, while that of B1C is 1.023 Mcps [7,8].

By referring to the official spatial signal interface document (ICD) of the BeiDou system, the B1C signal is composed of data component and pilot component [8]. The carrier center frequency of both is 1575.42 MHz, where the GPS L1 and Galileo E1 are also broadcasted on it. The structural characteristics of B1C can be listed in Table 1.

**Table 1.** The characteristics of the B1C signal.

| Signal Component | Center Frequency/(MHz) | Modulation | Rate Speed/(sps) |
|---|---|---|---|
| B1C_data | 1575.42 | BOC (1,1) | 100 |
| B1C_pilot | 1575.42 | QMBOC (6,1,4/33) | 0 |

The pilot component adopts quadrature multiplexed binary offset carrier (QMBOC) modulation, which allows the signal to be composed of a narrow-band component with larger power distribution and a broadband component with smaller power distribution [30]. This shows that the B1C signal can be compatible with high-performance receiving mode and low-complexity receiving mode, which is a relatively novel feature of the BDS-3 B1C signal.

The complex envelope form of B1C signal [31] can be written as:

$$S_{(B1C)} = s_{(B1C_D)}(t) + js_{(B1C_P)}(t) \tag{1}$$

where $s_{(B1C_D)}(t)$ is the data component of the signal, which is generated through the modulation of navigation message $D_{(B1C_D)}(t)$, ranging code $C_{(B1C_D)}(t)$, and subcarrier signal. $s_{(B1C_P)}(t)$ is the pilot component of the signal, which is modulated by subcarrier and ranging code $C_{(B1C_P)}(t)$, the navigation message is not modulated in it. $s_{(B1C_D)}(t)$ and $s_{(B1C_P)}(t)$ are distributed in the power ratio of 1:3. The modulated B1C signal can finally be written as the following expression in bandpass form [31]:

$$S_{(B1C)}^{(i)}(t) = \sqrt{2P_{(B1C)}}\left[\frac{1}{2}D_{(B1C)}^{(i)}(t)C_{(B1C_D)}^{(i)}(t)\cos(2\pi ft) + \frac{\sqrt{3}}{2}C_{(B1C_P)}^{(i)}(t)\sin(2\pi ft)\right] \tag{2}$$

In the formula, $S_{(B1C)}^{(i)}(t)$ represents the signal from the satellite labeled i, $D_{(B1C)}^{(i)}(t)$ represents the modulated navigation message, $P_{(B1C)}$ represents the signal power of B1C,

$C_{(B1C_D)}^{(i)}(t)$ is the ranging code of the data component, and $C_{(B1C_P)}^{(i)}(t)$ is the ranging code of the pilot component. The design of this data component and pilot component will greatly improve the ranging ability of the signal.

As for the ranging code, it is a layered code structure of B1C that the sub-code chip is strictly aligned with the time of the first code chip of the main code. The code rate of the B1C ranging code is 1.023 Mbps. It is obtained by truncating a Weil code with a length of 10243 chips. The Weil code sequence with length n is defined as follows:

$$W(t, w) = L(t) \oplus L(t + w) \tag{3}$$

$L(t)$ is a Legendre sequence of fixed length N, w is the phase difference between two Legendre sequences, and the value is 1 to 5121. The Legendre sequences of length N are defined as follows:

$$L(t) = \begin{cases} 1; & t \neq 0, \mathrm{int}x; \ t = x^2 \mathrm{mod} N \\ 0; & t = 0 \\ 0; & \mathrm{else} \end{cases} \tag{4}$$

The B1C ranging code can be obtained by cyclic interception of the Weil code.

B2a is another public navigation service signal broadcast by BDS-3. The BPSK(10) method is selected for its signal modulation in the data component and pilot component. Similarly, according to the spatial interface document of the BDS, we can express the signal in the form of a complex envelope [32]:

$$S_{(B2a)} = s_{(B2a_D)}(t) + js_{(B2a_P)}(t) \tag{5}$$

where $s_{(B2a_D)}(t)$ represents the data component of the signal, and $s_{(B2a_P)}(t)$ represents the pilot component of the data. We can also write the signal in bandpass form [32]:

$$S_{(B2a)}^{(i)}(t) = \sqrt{2P_{(B2a)}} \left[ D_{(B2a)}^{(i)}(t) C_{(B2a_D)}^{(i)}(t) \cos(2\pi ft) - C_{(B2a_P)}^{(i)}(t) \sin(2\pi ft) \right] \tag{6}$$

In this formula, $D_{(B2a)}^{(i)}(t)$ represents the modulated navigation message data, $C_{(B2a_D)}^{(i)}(t)$ is the data code of the data component, and $C_{(B2a_P)}^{(i)}(t)$ is the data code of the pilot component. $P_{(B2a)}$ gives the power of the B2a signal. The code rate of B2a ranging code is 10.23 Mcps and the code length is 10,230. Two thirteen-level linear shift registers $g_{1(x)}$ and $g_{2(x)}$ are used to generate two extended gold codes of 10,230 chips, and both of them obtain B2a ranging code through modulo-2 addition. The two polynomials are:

$$g_1(x) = 1 + x + x^5 + x^{11} + x^{13} \tag{7}$$

$$g_2(x) = 1 + x^3 + x^5 + + x^9 + x^{11} + x^{12} + x^{13} \tag{8}$$

for the data component and are:

$$g_1(x) = 1 + x^3 + x^6 + x^7 + x^{13} \tag{9}$$

$$g_2(x) = 1 + x + x^5 + + x^7 + x^8 + x^{12} + x^{13} \tag{10}$$

for the pilot component.

For the B2 band of BDS-3 new signal, it also broadcasts B2b signals. Both signals are composed of two orthogonal components. Therefore, the asymmetric constant envelope binary offset carrier technology (ACE-BOC) can be used to synthesize the two signals to form a constant envelope signal, which can reduce the payload and multiplexing loss. The use of ACE-BOC makes the signal receiving of the B2 signal highly flexible: the above two signals in the B2 band can be received as broadband signals or two separate quadrature phase-shift keying (QPSK) signals.

### 2.2. Principles in Zero-Baseline Experiments

Zero-baseline means that the antenna phase centers corresponding to multiple signal channels are consistent, i.e., the same antenna is used to receive signals. In the study, the GNSS signal recorder used in the zero-baseline experiment has four signal input channels corresponding to different frequencies and signal branches. As shown in Figure 2, after the signal is led out from the antenna, the power distributor is used to connect the signal to the four signal ports through the same transmission conditions, and the zero-baseline condition is formed between the four signal receiving terminals. For the actual recorder, the signal channel 1 and signal channel 2 will operate on the same frequency point, and signal channel 3 and signal channel 4 will operate on another frequency point.

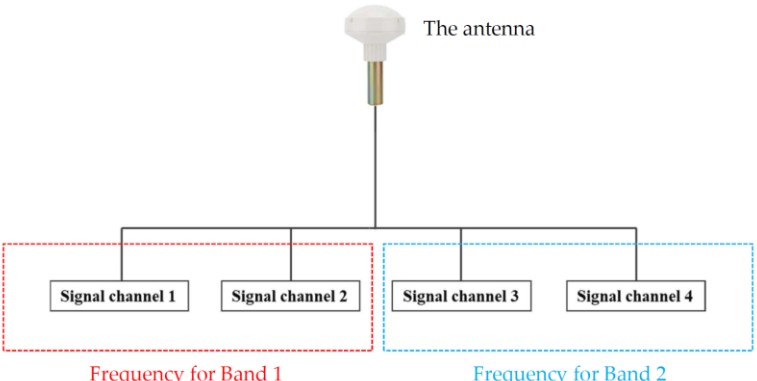

**Figure 2.** Diagram of the zero-baseline experiment in this study.

Since the four signal channels were connected with the same antenna, the error influence is consistent among the four signal channels in the process of the GNSS signal transmission through satellite, ionosphere, and atmosphere propagation and the signal line before obtaining access to the recorder. Therefore, theoretically, the data collected by signals of the same signal frequency between different signal channels should be consistent. However, in practice, owing to the influence of the structure of the ranging code itself, the performance of the ranging code cannot be fully idealized. It leads to certain differences in the ranging code data of different signal channels at the same epoch and frequency, which can be used to reflect the ranging performance of the signal frequency ranging code directly.

The code signal path differences can quantitatively reflect the difference in the ranging code data. In order to obtain the values of code signal path differences, the code-level waveform diagram of the two signal channels were obtained by phase correlation between local codes and two-channel signals. Figure 3 shows the code-level waveform diagram obtained after two signal channels pass through the same epoch. Generally, the waveform figure at a single epoch has a power maximum near the central position, where the path delay is 0 and the power on both sides decreases with the increase in the absolute value of the path. Delay difference between the extreme positions of the waveform between two channels code signal path differences of two signal channels. In the observation period, the code signal path differences of each epoch can be obtained through the same data process under different epochs that meet the observation requirements, and the changing image of code signal path differences with time can be obtained.

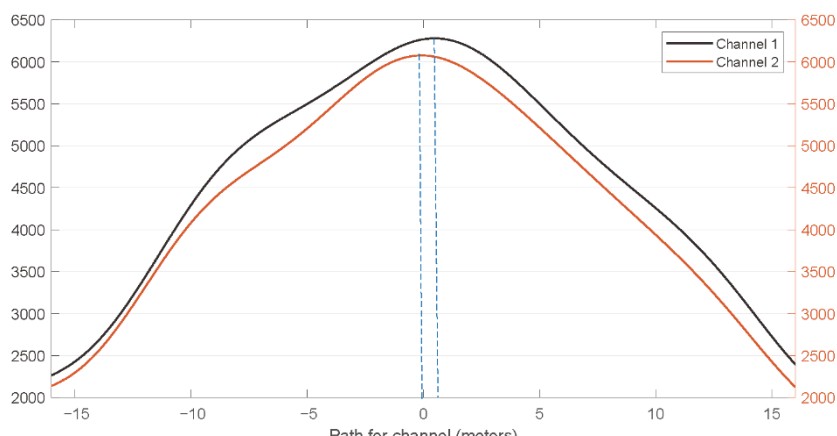

**Figure 3.** The calculation of code signal path differences. Code signal path differences are the path difference between the two blue lines in the figure.

Ideally, the code signal path difference of the waveform signals in the same epoch should be strictly 0 due to the zero-baseline environment. Therefore, the code signal path differences can be used to evaluate the performance and ranging accuracy of the ranging code directly. The smaller the code signal path differences, the better the performance and ranging accuracy of its corresponding ranging code. In order to obtain the code-level signal path differences for each signal channel, we use a 4-channel IF signal recorder corresponding to Figure 2 that can receive B2 and B1 band signals simultaneously to collect the raw IF signal. There is two signal receiving channels under each signal frequency point. The collected raw IF signal is processed by a self-developed SDR based on MATLAB to obtain the code signal path differences of two signal input channels at the same frequency with a zero-baseline. The raw IF data processing of GNSS signal recorder under a single channel can be shown in Figure 4.

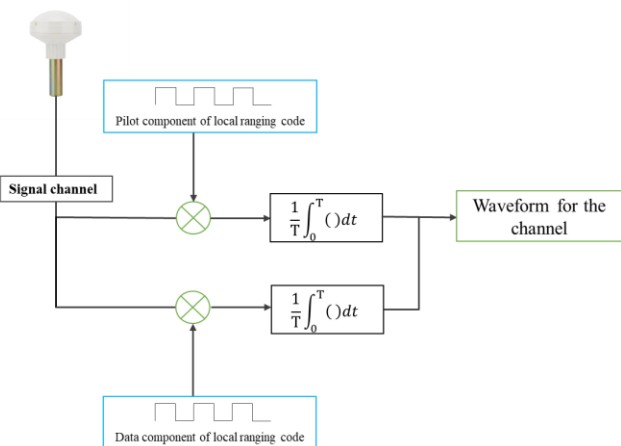

**Figure 4.** The IF data acquisition and process for one channel.

After the signal is transmitted to an input port through a four-power divider, it is first divided into two sub-channels inside the receiver channel for further processing. SDR will process the data epoch by epoch. One sub-channel of the signal is cross-correlated with the pilot component signal of the local code, and the other sub-channel of the signal is cross-correlated with the data component signal of the local code. A set of waveform data are obtained from the coherent integration of two signals for a certain epoch. According to the ICD files of B1C and B2a, the waveform data of the pilot component and the data component are superimposed coherently according to the power ratio of the photographic frequency point to obtain the waveform data under this channel.

### 3. The Zero-Baseline Experiment

Our experiment was conducted on the roof of a high-rise building on the campus of Shandong University (Weihai), Weihai, Shandong Province, China. The surrounding environment of the experiment is open, which is convenient for conducting zero-baseline experiments. Our experiment was conducted on 7 March 2022. The day was sunny, and the weather conditions remained stable throughout the day, which was conducive to the operation of the zero-baseline experiment. The data receiving and acquisition system used in the zero-baseline experiment is mainly composed of a coking-ring GNSS antenna and a four-channel IF signal recorder. Figure 5a,b record the field environment of the zero-baseline experiment.

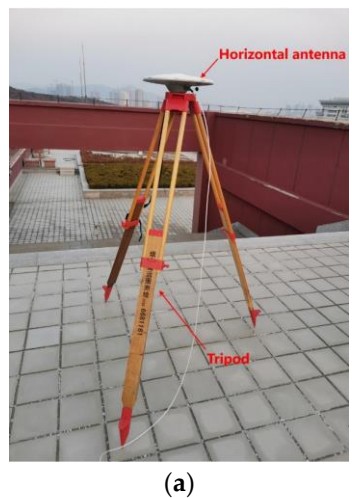
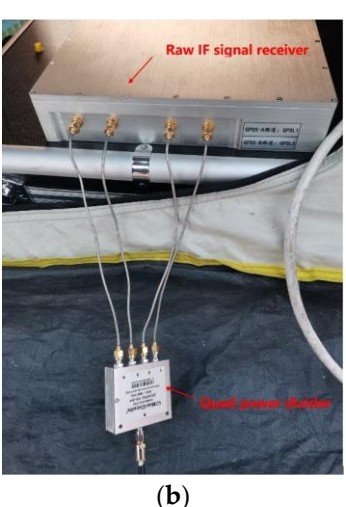

(**a**)          (**b**)

**Figure 5.** The field of the zero-baseline experiment: (**a**) Horizontally placed geodetic GNSS antenna and its tripod; (**b**) Quad-power divider and raw IF signal recorder.

In Figure 5a, the geodetic GNSS antenna is erected and fixed by a tripod to keep it horizontal. The signal is led out from the antenna by one shielded cable. It can be noted that there is a fence on the top of the building, the horizontal plane of the erected antenna is higher than the height of the fence, and the other positions remain open. Therefore, it can be considered that the antenna signal is not affected by the surrounding obstacles. Figure 5b shows the wiring of the signal receiving and processing terminal. The shielded cable is led out from the antenna and connected to the four power dividers in Figure 5b. Before entering the recorder, the signal is divided into the same four branches by the four power dividers and connected to the four signal receiving ports of the recorder, respectively. It should be noted that the four data lines connecting the power divider and the recorder should be completely consistent to eliminate the difference in the line's thermal noise. The bandwidth of the recorder is 20.46 MHz, of which two signal channels will receive the B1 band with a frequency center of 1529 MHz, which can collect B1C signals with the center frequency of 1575.42 MHz. The other two channels will receive the B2 band with a frequency center of 1130 MHz, which can collect B2a signals with the center frequency of 1176.45 MHz. In Figure 5, the two input interfaces on the right correspond to the B1 band of the signal. It can be considered as the signal channel 1 and signal channel 2 of Figure 2. The two input interfaces on the left correspond to the B2 band of the signal. They can be considered as signal channel 3 and signal channel 4 of Figure 2. The recorder works at 62 MHz continuous sampling, and the collected IF data are quantized by 2 bits. After the signal is processed by the original IF signal receiver, the original IF signal data are stored in the data hard disk connected to the receiver at an interval of two minutes for subsequent data processing. In addition, in the later data processing, in order to minimize the impact of the multipath, the data with satellite elevation above 30 degrees were selected

for processing. Thanks to the open surrounding environment during the experiment, the data can be arbitrarily selected from the satellite azimuth of 0 to 360 degrees.

## 4. Results

### 4.1. Satellite and the Acquisition of Data

The time range of the experimental data collection is 11:02 to 18:00 on 7 March 2022 (Beijing time). Data processing and analysis shall be carried out within the time range when the satellite elevation meets the requirements. Our selected satellite elevation mainly ranges from $30°$ to $70°$, and the satellite elevation can be calculated from the satellite's broadcast ephemeris. The duration of raw IF data used for the data processing shall be greater than 1 h to ensure the effectiveness of the analysis results. When selecting the period, we try to ensure that the elevation of the selected satellite is within a certain similar range, so as to reduce the impact of satellite elevation when analyzing the ranging accuracy alone under a specific PRN. At the same time, considering the slow processing speed of SDR and the huge storage space required for data results, we basically choose an interval of 1 to 3 h for further processing, which not only ensures that the satellite elevation conditions are met, but also ensure the processing process is not too long. As the structure of the ranging code broadcasted from a satellite system is similar under a frequency point, the ranging performance of the system can be reflected by processing at least one set of satellite data, and the conclusion is not affected by the number of satellites and satellite orbit type. Therefore, only limited satellite data have been processed and analyzed in our research.

The signals from satellites with PRN 37 (MEO) and PRN 39 (IGSO) were used for the ranging accuracy analysis of B2a. The satellite with PRN 37 is an MEO satellite. Its satellite elevation has a large variation range, so we can also collect and process its data for a long time to study the relationship between ranging accuracy and the satellite elevation. When processing B1C frequency points, we use satellite PRN 39 for data processing, which is consistent with the selection of B2a satellites. This is an IGSO satellite, which can ensure a high satellite elevation and a long time in the observation period. At the same time, thanks to the good interoperability and compatibility of BDS-3, the B2a is consistent with the center frequency of GPS L5C. Therefore, we use the same raw IF signal to analyze the ranging accuracy of L5C PRN 4 for comparison with the BDS-3. GPS PRN 4 satellite is also an MEO satellite, which has a large range of elevation changes. We can use this satellite to compare the ranging performance of the GPS L5 with that of the BDS-3 frequency point of B2a and B1C.

In the experiment, the zero-baseline experimental environment can only ensure that all conditions are consistent before the signal is connected to each channel of the IF signal recorder. There are four channels and two crystal oscillators in our recorder. Two of the channels share one crystal oscillator and a frequency point, while four channels share a clock. So, the bias between the channels at the same frequency point is small. In our work, the path difference between two channels at the same frequency point are calculated to investigate the accuracy of the ranging code. Therefore, in the data processing, we ignore the system error of the receiver and consider the hardware delay equal in each signal channel. We calculate the code signal path differences directly after mapping the waveform figure as Figure 3. By analyzing the solutions in Table 2, we find that there is no bias for the two channels at the frequency of 1176.45 MHz, while there is a 26 cm bias for the two channels at the frequency of 1575.42 MHz. When signals of all channels are processed by the GNSS-SDR according to Figure 4, the coherence integration time is set as 10 ms to enhance the signal-to-noise ratio. After processing 10ms data, the complete waveform data can be obtained, and the code signal path differences can be calculated by the differences between the extreme values of the waveform. As the calculation efficiency of GNSS-SDR is relatively low, we conduct data calculation at an interval of 50 ms, and 19 waveform data will be obtained in a single channel within 1 s. After the code signal path differences calculation of each of the 19 waveform data, the median value is taken as the code signal path differences value in this second. The standard deviation of 19 groups of

data within 1 s can also be calculated accordingly, which reflects the dispersion degree of internal code signal path differences in 1 s, and its satellite elevation-varying figures can also be given. By arranging the results of each second in the processing period, we can obtain the corresponding time sequence change diagram.

**Table 2.** Statistical characteristics of code delay.

| PRN | GPS L5C | BDS B2a | | BDS B1C |
|---|---|---|---|---|
| | **4** | **37** | **39** | **39** |
| RMS(m) | 0.1603 | 0.1526 | 0.1527 | 0.2674 |
| STD(m) | 0.1585 | 0.1309 | 0.1485 | 0.1779 |
| Mean(m) | −0.0234 | −0.0785 | −0.0358 | 0.1996 |

### 4.2. Influence of the Change of Satellite Elevation

Firstly, we analyze the influence of the satellite's high elevation on the ranging accuracy of a single frequency point. As mentioned above, the BDS-3 satellite of PRN 37 is an MEO satellite, which satellite elevation changes greatly. We use the SDR of the B2a frequency point to process the raw IF data of PRN 37. The selected processing period is 13:09–17:23 (Beijing time). During this period, the satellite elevation gradually increased from 30° to the maximum, and then gradually decreased back to 30 degrees. According to Section 4.1, the code signal path differences of B2a and the satellite elevation time series change diagram can be seen in Figure 6.

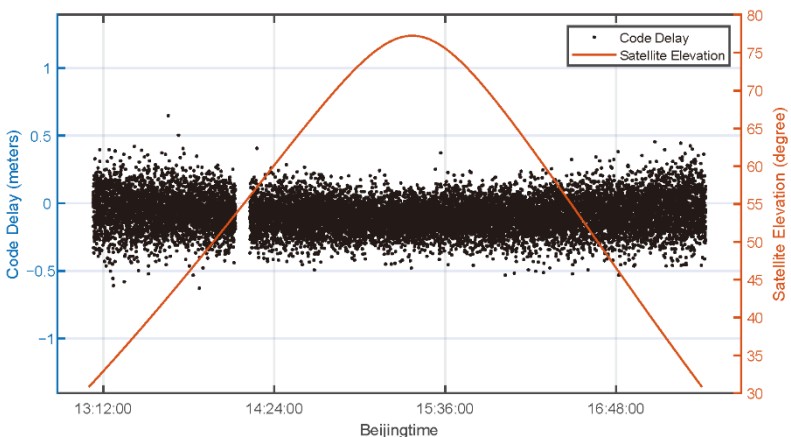

**Figure 6.** Time distribution of B2a PRN 37 satellite code signal path differences and satellite elevation.

It should be noted that the time starting point and length of the data processing period selected at each frequency point is different. As described in Section 4.1, this is to meet the requirements of satellite elevation similarity. In addition, the processing speed of SDR is slow. Choosing 1–3 h instead of the whole range of data for processing can improve the processing efficiency and ensure the accuracy of the results.

It should be noted that there are certain data interruptions in Figure 6, which are empty data records caused by loose data connection ports. It can be preliminarily seen from the Figure 6 that the absolute value of the code signal path differences remains within 0.5 m during the observation period. The code signal path differences converge towards 0 m in the state of high satellite elevation, and they are relatively discrete in the state of low satellite elevation. Standard deviation (STD) can be used to quantitatively describe the degree of data dispersion, which is an indicator of ranging accuracy. In the process of processing, since 19 code signal path differences will be recorded in one second, the STD of these 19 data in one second can reflect the dispersion of the code signal path differences in that second. The same processing is performed on the data in the selected period to obtain

all the STD distribution data at the 1 s interval. STD data with satellite elevation are plotted as shown in Figure 7.

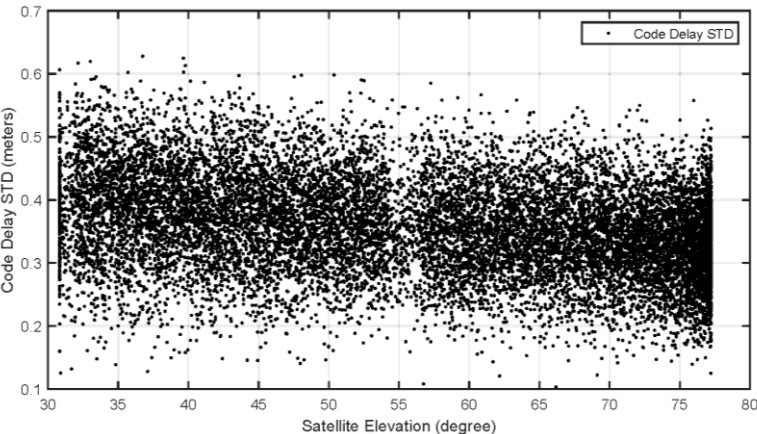

**Figure 7.** Satellite elevation distribution of B2a PRN 37 satellite code signal path differences STD.

In Figure 7, with the increase in satellite elevation, it can be seen that the STD value has a very obvious downward trend. It is noted that there are many data points recorded at high satellite elevation, which is caused by the slow change in satellite elevation, and a large number of data points are recorded when the satellite operates at high elevation. The relationship between STD and satellite elevation will be described quantitatively. According to the data in Figure 7, the data correlation analysis can be carried out. The Pearson correlation coefficient matrix is used to describe this quantitative correlation. The STD value and satellite elevation can be regarded as two variables, and the relationship between them can be described by a two-dimensional matrix P:

$$P = \begin{bmatrix} 1 & -0.308 \\ -0.308 & 1 \end{bmatrix} \tag{11}$$

In this matrix P, the main diagonal element represents the correlation coefficient of the two variables; that is, 1 represents a complete correlation. The sub-diagonal represents the Pearson correlation coefficient between STD value and satellite elevation. The closer the absolute value of the Pearson correlation coefficient is to 1, the stronger the correlation between the two variables they have. The Pearson correlation coefficient between STD and satellite elevation is $-0.308$. Therefore, we can conclude that there is a weak negative correlation between the satellite elevation and the STD value; this conclusion is only satisfied in this paper when the satellite elevation is between $30°$ and $90°$. The period with high satellite elevation will produce better ranging accuracy, but the accuracy changes slightly with the satellite elevation.

The higher the satellite elevation, the greater the signal power and the lesser the multipath effect. So, the high satellite altitude angle has a smaller code signal path difference. Since the satellite altitude angle only affects the code signal path differences of a single satellite within a period and has nothing to do with the signal structure differences between different frequency points. We can draw the above conclusions by processing only one satellite for each frequency point.

*4.3. Ranging Accuracy Analysis*

In this subsection, the ranging accuracies of BDS-3 B2a and B1C according to code signal path differences are analyzed. We processed the measurements from PRN 39 for B2a from 11:05 to 13:40 and PRN 39 for B1C from 12:00 to 13:15, while plotting the code signal path differences and the time distribution of satellite elevation. In addition, the center frequency of the GPS L5C frequency point is the same as that of BDS B2a. We processed its data from 11:05 to 13:09 for comparisons. The corresponding calculation results are shown

in Figures 8–10. Figure 8 shows the calculation results of B2a PRN39 satellite; Figure 9 shows the calculation results of B1C PRN39 satellite; Figure 10 shows the calculation results of L5C PRN4 satellite.

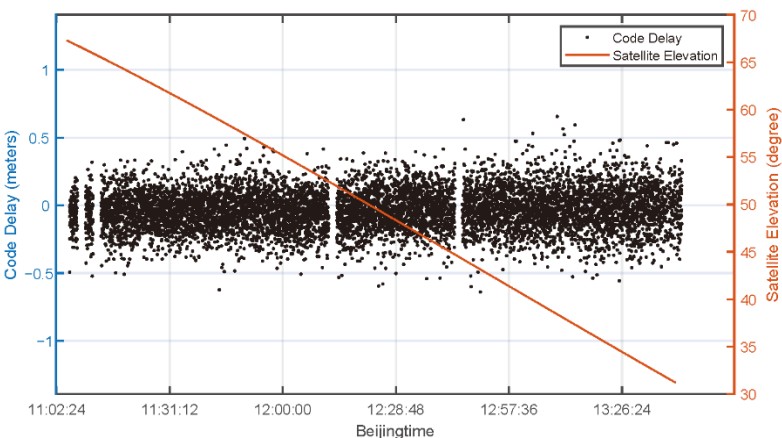

**Figure 8.** Time distribution of B2a PRN 39 satellite code signal path differences and satellite elevation.

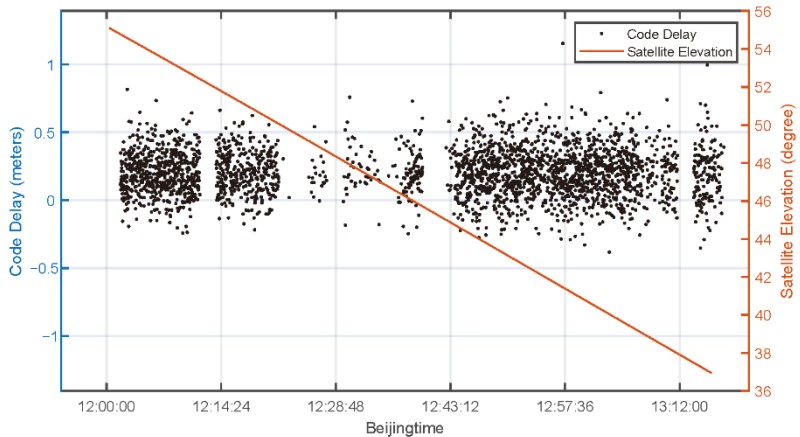

**Figure 9.** Time distribution of B1C PRN 39 satellite code signal path differences and satellite elevation.

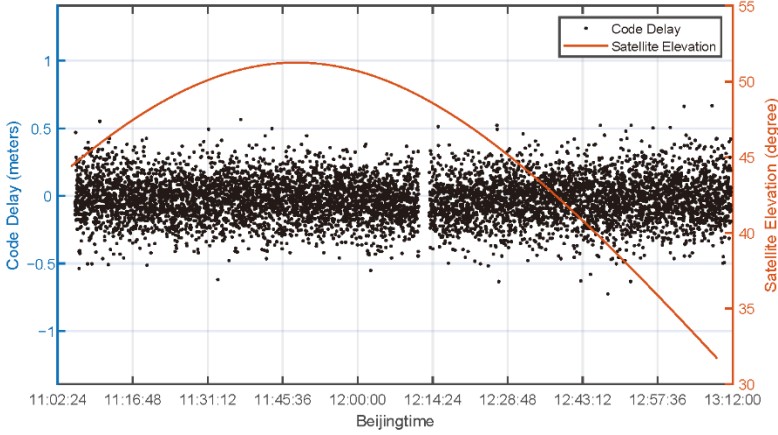

**Figure 10.** Time distribution of L5C PRN 4 satellite code signal path differences and satellite elevation.

It is noted that when processing the data of B1C, the figure is relatively loose, which is mainly caused by two reasons below. One is that the loose data interface makes certain empty data records exist during data sampling. On the other hand, the data with code signal path differences greater than 0.8 m or greater than three times RMS is considered as gross error and removed. The distribution of these data is messy and not the main component of

the data composition, so they should be regarded as gross errors and discarded. In addition, the data observation duration of B1C is about 1 h shorter than that of B2a. This is because the data point processing results in the selected period of B1C are denser than those in other periods. The variation range of the satellite elevation data is similar to that of B2a, which can more intuitively reflect the distribution of code signal path differences. The observation duration meets the minimum 1 h requirement described in Section 4.1. In this way, the overall characteristics of B1C signal code signal path differences can be seen during this period of time, and the decline in the amount of data will not affect the main conclusions. It is worth noting that there is an obvious overall data deviation in the calculation results of B1C. Except for the B1C signal, the code signal path difference of other signals is between −0.5 m and 0.5 m, which means that their ranging accuracy remains basically unchanged. However, compared with other analyzed frequency points, the calculation result of B1C has a very significant deviation from the theoretical value 0 m, and the whole calculation result shows an upward trend. To quantitatively describe this change, the overall code signal path differences RMS, mean value, and STD of each frequency point in the observation processing time can also be calculated. The corresponding processed figures and data results are given in Table 2.

The RMS, STD, and mean value of satellite data in the whole period are calculated. It has been pointed out in Section 4.2 that there is only a weak negative correlation between satellite elevation and ranging accuracy when the satellite elevation is between 30° and 90° in our study. Therefore, the influence of satellite elevation can be ignored if the results of long-term observation are used. Moreover, the conclusion of ranging accuracy will not be affected by the number of satellites and processing time. It can be considered that the conclusions drawn from the satellites selected in this experiment are universal. Under the BDS B2a frequency point, the RMS values are 0.1526 m for PRN 37 and 0.1527 m for PRN 39, respectively. We note that BDS B2a covers two satellite orbit types, MEO and IGSO, and their data results are basically consistent. Therefore, the difference of satellite orbit types can be ignored, and it can be considered that the result of ranging accuracy is only related to the signal structure of different frequency point. Under the BDS B1C frequency point, our RMS value is 0.2674. We can note that the RMS value of B1C is lower than the RMS value of B2a. To enhance the comparison of data, we use the same raw IF data and processing method to calculate the code signal path differences of the GPS L5C band. The results show that the ranging accuracy of BDS-3 B2a and GPS L5C are roughly the same, and both of them are significantly better than the ranging accuracy of B1C. For the average value of code signal path differences, the calculation results of GPS L5C and BDS B2a are on the order of $10^{-2}$ m, while the average value of the B1C signal is 0.1996 m, which is far from the theoretical value of zero. The experimental data also show that B2a and GPS L5C have similar ranging performances. The code rate and code length of the GPS L5C signal are consistent with those of B2a, so it has a similar ranging performance, which is mutually confirmed with the experimental results. Therefore, this zero-baseline experiment has certain reliability. In addition, the standard deviation of B1C code signal path differences are also significantly greater than that of other frequency points. These results show that B1C has poor ranging accuracy in the zero-baseline experiment. We try to explain this from the difference in signal structure.

The particularity of the B1C signal structure may cause some deviation in the ranging results. In Section 2.1, we know that the pilot component of the B1C signal is modulated by QMBOC. In fact, the QMBOC subcarrier is composed of two bipolar subcarriers, so the B1C signal can be regarded as a combination of three bipolar components [31]:

$$
\begin{aligned}
S_{(B1C)}^{(i)}(t) = {} & \tfrac{1}{2} D_{(B1C)}^{(i)}(t) C_{(B1C_D)}^{(i)}(t) \cdot \mathrm{sign}(\sin(2\pi f_a t)) \\
& + \sqrt{\tfrac{1}{11}} C_{(B1C_P)}^{(i)}(t) \cdot \mathrm{sign}(\sin(2\pi f_b t)) \\
& + j\sqrt{\tfrac{29}{44}} C_{(B1C_P)}^{(i)}(t) \cdot \mathrm{sign}(\sin(2\pi f_a t))
\end{aligned}
\tag{12}
$$

where $f_a = 1.023$ MHz and $f_b = 6.138$ MHz. In some research, the effect of the subcarrier is interpreted as a selective communication channel that distorts the useful signal [33]. The existence of subcarriers may affect the accuracy of ranging code at SDR reception. For ranging codes, B1C and B2a have the same code length, but the code rate of B2a is ten times that of B1C. A lower code rate may affect the accuracy of signal ranging. In this study, the coherence time of B2a and B1C is 10 ms. B1C signal has wider bandwidth. Although the wide bandwidth of B1C is conducive to combating multipath effects, the existence of subcarriers and lower code rate causes the positioning result to shift. Affected by the above factors, B1C has lower ranging accuracy in this zero-baseline experiment. Although some studies indicate that the ranging accuracy of B2a is lower [34], its testing environment should be affected by adverse environmental factors, which needs further discussion. We will conduct a more detailed study on this issue in future work.

## 5. Discussion and Conclusions

The zero-baseline experiments were used to evaluate the GNSS positioning performance. This paper presents a method to figure out the accuracies of BDS-3 new civil ranging codes directly using the data collected from a zero-baseline experiment. The results show that the B2a code ranging accuracy is better than that of B1C, and similarly with GPS L5C code.

About 7 h of raw IF data were collected on two frequencies. For a certain frequency point, at least one hour of data were selected for processing. The final calculation results show that the precision of the new civil signal B1C is in correspondence with B2a, while B1C has a larger STD value. The difference between B2a and B1C signal structures has a great impact on the ranging accuracy. B2a has a higher code rate, so it has higher ranging accuracy. Although B1C has a wide bandwidth, its accuracy is reduced due to its slow code rate. At the same time, compared with GPS signals, it can be found that B2a and L5C have similar ranging accuracy, owing to the same code rate and code length. In addition, the dispersion of the ranging results has weak negative correlation between satellite elevations when the satellite elevation meets $30°$ to $90°$, for which the Pearson correlation coefficient is $-0.308$. Under the condition of high satellite elevation, the data onto one second have a relatively small STD value, that is, higher ranging accuracy.

In this work, the amount of the raw IF data storage is too large, and the speed of processing the data using the GNSS-SDR software receiver is too slow. We only processed and analyzed parts of the data in the environment with good weather and an open surrounding area. The analysis of this work only involves the B2a and B1C signals of BDS-3 with the L5 signal of GPS. Therefore, we also plan to compare the ranging performance of each frequency band signal of other GNSSs in future work to expand the universality of the conclusion.

**Author Contributions:** Conceptualization, Y.L. (Yu Liu) and F.G.; data curation, Y.L. (Yu Liu), J.L., Y.L. (Yang Liu) and Y.Q.; formal analysis, Y.L. (Yu Liu); funding acquisition, F.G.; investigation, Y.L. (Yu Liu), J.L., B.N., Y.L. (Yang Liu) and Y.Q.; methodology, Y.L. (Yu Liu); project administration, F.G.; resources, Y.L. (Yu Liu); software, Y.L. (Yu Liu), J.L., B.N. and S.C.; supervision, F.G.; validation, Y.L. (Yu Liu), J.L., Y.L. (Yang Liu) and Y.Q.; visualization, J.L.; writing—original draft, Y.L. (Yu Liu); writing—review and editing, F.G., J.L. and Y.H. All authors have read and agreed to the published version of the manuscript.

**Funding:** This research was funded by Wenhai Program of the S&T Fund of Shandong Province for Pilot National Laboratory for Marine Science and Technology (Qingdao) (NO.2021WHZZB1004) and the Program of the National Natural Science Foundation of China, (grant number 41604003, 41704017).

**Data Availability Statement:** The data used in the experiment are managed by the Institute of Space Science of Shandong University, and the author can be contacted to obtain the experimental data.

**Acknowledgments:** The author thanks the satellite navigation and remote sensing research group of the Institute of Space Science of Shandong University for its site and technical support.

**Conflicts of Interest:** The authors declare no conflict of interest.

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
