# Peer review of "Analysis and Performance Evaluation of BDS-3 Code Ranging Accuracy Based on Raw IF Data from a Zero-Baseline Experiment"

_remotesensing, doi:10.3390/rs14153698_

Round 1

Reviewer 1 Report

Ramarks:

1. The reasons for selecting 2-3 hour intervals from the full 7 hours of observations were not described. Why are these time intervals different for different frequency points?

2. A great deal of observational data has been rejected for BDS-3 B1C PRN39 (outliers). Such a large number of outliers has a reason that is not explained in any way in the manuscript. Including these observations in your calculations will completely change the results.

3. Why were the accuracy analyzes for the B1C code of the BDS-3 system performed only for the IGSO satellite and not for the MEO satellite? The results of the comparison with the GPS satellite (MEO) are therefore, in my opinion, not very convincing.

4. "… calculation result of B1C has shifted by 0m, showing an upward trend" (line 419). A shift of 0m means no shift.

Author Response

Dr.Reviewer,

We have carefully revised the manuscript according to your comments, and we have given corresponding replies to your review comments. Please see the attachment.

Reviewer 2 Report

The paper deals with an interesting and active research topic, namely the assessment of BDS-3 signals’ performance. Moreover, the authors use zero-baseline and a software defined receiver (SDR) for their investigations, which is a quite new approach. The paper is generally well structured and prepared. The usage of English is good and the title describes well the study. However, in order to improve the article, some points should be addressed, as described in the following:

Experiment design
The authors use an SDR for their investigations. This is an alternative approach to the use of hardware receivers. The authors should describe the two approaches and outline the advantages and disadvantages of each approach. (E.g. in section 4.1 the limited processing capacity is mentioned as a disadvantage. What about the advantages?) I suggest to devote a small paragraph to that (for example in the introduction) and add a sentence also in the abstract.

In section 2.2. it is stated “Ideally, the code signal path difference of waveform signals in the same epoch should be strictly 0 due to the zero-baseline environment. Therefore, the code signal path differences can be used for evaluate the performance and ranging accuracy of ranging code directly.” It is not clear how the 4 signal channels are realized. Is this done using any hardware for separating the channels for the different frequency bands shown in Figure 2 (and may be also for the channels within the same band)? If a hardware/device is used, it could introduce a bias between the channels. In lines 319-322 the authors mention that “Before the zero-baseline experimental, we checked the recorder to confirm that the noise between its internal channels fluctuates within a reasonable range. Therefore, in the data processing, we ignore the system error of the receiver and consider the hardware delay equal in each signal channel”. The noise level can be the same on two channels, however this does not ensure that there is no differential delay (bias) between the channels. The authors should address the possibility of any bias between the channels (describe precisely how the channels are realized, how they checked the noise level and how can we be sure that no inter-channel bias exists).

Conclusions
The authors are stating that “there is only a weak negative correlation between satellite elevation and ranging accuracy”. This is stated e.g. in line 426 but also in other places in the text. This is true for the elevation ranges tested in the study (above 30 degrees). This condition should be included in the statements (e.g. for the elevation range studied here (30-90 degrees …) because at lower elevations the correlation becomes really high. 

Typos and other corrections
There are several minor corrections to be done, as described in detail in the attached file.

Final recommendation
The paper can be considered for publication, after minor revisions.

Author Response

(The authors gave the same response as above.)

Reviewer 3 Report

Corrections are given in the PDF document.

Author Response

(The authors gave the same response as above.)
